# GraphSearch: An Agentic Deep Searching Workflow for Graph Retrieval-Augmented Generation

## Abstract

Graph Retrieval-Augmented Generation (GraphRAG) enhances factual reasoning in LLMs by structurally modeling knowledge through graph-based representations. However, existing GraphRAG approaches face two core limitations: shallow retrieval that fails to surface all critical evidence, and inefficient utilization of pre-constructed structural graph data, which hinders effective reasoning from complex queries. To address these challenges, we propose GRAPHSEARCH, a novel agentic deep searching workflow with dual-channel retrieval for GraphRAG. GRAPHSEARCH organizes the retrieval process into a modular framework comprising six modules, enabling multi-turn interactions and iterative reasoning. Furthermore, GRAPHSEARCH adopts a dual-channel retrieval strategy that issues semantic queries over chunk-based text data and relational queries over structural graph data, enabling comprehensive utilization of both modalities and their complementary strengths. Experimental results across six multi-hop RAG benchmarks demonstrate that GRAPHSEARCH consistently improves answer accuracy and generation quality over the traditional strategy, confirming GRAPHSEARCH as a promising direction for advancing graph retrieval-augmented generation.

## 1 Introduction

Large Language Models (LLMs) demonstrates remarkable capabilities in natural language understanding and reasoning (Zhao et al., 2023; Naveed et al., 2025). Despite their strong performance, LLMs inherently rely on their parametric knowledge, which often results in hallucinations and a lack of factual grounding (Zhang et al., 2025; Wang et al., 2023). Retrieval-augmented generation (RAG) has emerged as a paradigm that combines LLMs with external knowledge bases, enhancing factuality, credibility and interpretability in knowledge-intensive tasks (Lewis et al., 2020).

More recently, Graph Retrieval-Augmented Generation (GraphRAG) is introduced to overcome the shortcomings of traditional RAG, which relies solely on semantic similarity for retrieval (Peng et al., 2024). By constructing

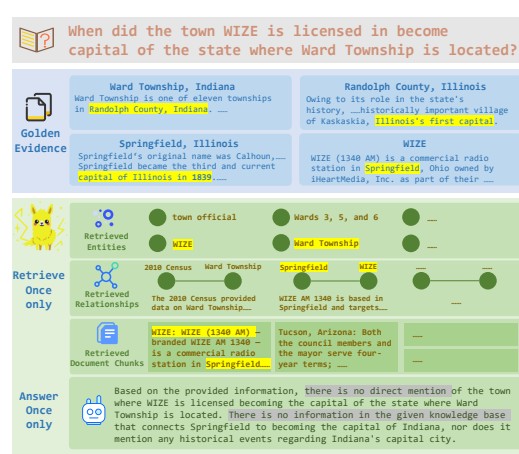

Figure 1: Shallow retrieval in GraphRAG.

structural graph knowledge bases (graph KBs) and leveraging hierarchical retrieval strategies, GraphRAG strengthens the integration of contextual information across massive entities and relationships (Sarthi et al., 2024; Edge et al., 2024; Guo et al., 2024). Building upon this foundation, some advanced graph-based enhancements that incorporate diverse structures, including heterogeneous graphs, causal graphs, and hypergraphs, to enrich representational ability and facilitate more abundant graph construction (Fan et al., 2025; Wang et al., 2025; Luo et al., 2025; Feng et al., 2025; Xu et al., 2025). In addition, heuristic strategies such as path-based search, pruning, and

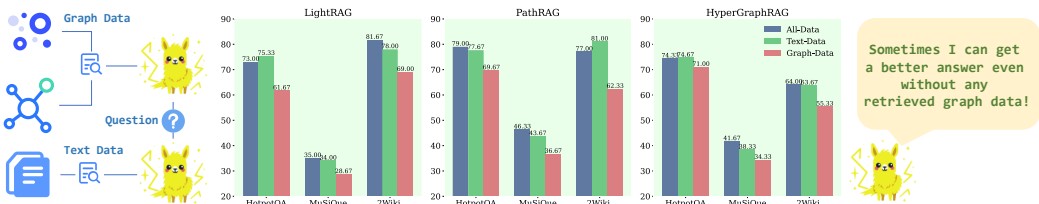

Figure 2: Comparison of using graph data only, text data only, or all data as commonly adopted in GraphRAG approaches. The metric is SubEM. The contribution of retrieved graph data is marginal.

memory-inspired indexing further reinforce reasoning abilities and enable deeper multi-step exploration (Chen et al., 2025; Jimenez Gutierrez et al., 2024; Gutiérrez et al., 2025; Wang, 2025).

However, existing GraphRAG approaches still face challenges that lead to performance bottlenecks: (i) **Shallow retrieval results in missing evidence for complex queries.** Most GraphRAG methods adopt a single-round retrieval-and-generation process as the interaction strategy between the LLM and the graph KB (Edge et al., 2024; Guo et al., 2024; Fan et al., 2025). However, as illustrated in Figure 1, when handling a complex query that requires four pieces of golden evidence, *"When did the town WIZE is licensed in become capital of the state where Ward Township is located?"*, the entity *Randolph County* is not retrieved by the LightRAG retriever. Consequently, the LLM's reasoning suffers from broken logic and insufficient evidence. (ii) **Limited ability to exploit structural data due to constrained retrieval scope**. Existing GraphRAG methods with heuristic path-construction schemes (Fan et al., 2025; Chen et al., 2025; Jimenez Gutierrez et al., 2024) often fail to fully leverage the structural information in graph KBs, fundamentally because shallow retrieval restricts the coverage of relevant nodes and relations. Without sufficient coverage of retrieved subgraphs, the available structural signals are fragmented and sparse, making it difficult for LLMs to integrate semantic and structural modalities for complex reasoning. As shown in Figure 2, models may perform comparably with text-only evidence, highlighting that the underutilization of graph data is tightly coupled with the limitations of current retrieval strategies.

We propose **GRAPHSEARCH, an agentic deep searching workflow for GraphRAG**. As illustrated in Figure 3, GRAPHSEARCH is a novel agent framework designed to access graph KBs through dual-channel retrieval, acquiring both semantic and structural information, and performing multi-turn interactions to complete complex reasoning tasks. Targeting the shallow retrieval problem in existing GraphRAG approaches, **GRAPHSEARCH models retrieval as a modular searching pipeline**, which consists of six modules: *Query Decomposition (QD)*, *Context Refinement (CR)*, *Query Grounding (QG)*, *Logic Drafting (LD)*, *Evidence Verification (EV)*, and *Query Expansion (QE)*. Through the coordinated contributions of these modules, GRAPHSEARCH decomposes complex queries into tractable atomic sub-queries, retrieves fine-grained knowledge from graph KBs, and iteratively performs logical reasoning and reflection to remedy missing evidence. Furthermore, **GRAPHSEARCH adopts a dual-channel retrieval strategy**, constructing semantic queries over chunk-based text data and relational queries over structural graph data, thereby fully synergizing both modalities and integrating them into contexts that support LLMs in complex reasoning.

We conduct experiments on six multi-hop RAG datasets. The results demonstrate that leveraging the graph KBs retrievers built upon the corresponding GraphRAG approaches, GRAPHSEARCH consistently outperforms the single-round interaction strategy in terms of answer accuracy and generation quality, while also exhibiting strong plug-and-play capability, as shown in Table 1. Furthermore, the effectiveness of the dual-channel retrieval strategy, the contributions of agentic modules, and its robustness under a small-scale LLM and varying retrieval budgets are all empirically validated.

Our contributions are as follows: (i) We propose GRAPHSEARCH, an agentic deep searching workflow that overcomes the challenges of shallow retrieval and the ineffective use of graph data in existing GraphRAG approaches. (ii) We introduce a modular searching pipeline with coordinated modules for iterative reasoning and a dual-channel retrieval strategy integrating semantic and relational queries over graph KBs. (iii) Experiment results across six multi-hop RAG datasets demonstrating that GRAPHSEARCH consistently outperforms vanilla GraphRAG in accuracy and quality.

## 2 RELATED WORK

### 2.1 GRAPH RETRIEVAL-AUGMENTED GENERATION

RAG augments LLMs with external evidence to improve factuality of knowledge-intensive tasks (Lewis et al., 2020). Building on this, GraphRAG is an advance paradigm that explicitly models structural relations among entities, thereby capturing relational semantics, contextual dependencies and structural knowledge integration (Peng et al., 2024; Edge et al., 2024). Early work (Sarthi et al., 2024; Edge et al., 2024) emphasize hierarchical summarization and global information integration, but they insufficiently leveraged the fine-grained structural information. LightRAG (Guo et al., 2024) advanced this direction by incorporating graph structures into both indexing and retrieval. Recent efforts in graph KB construction introduce diverse structural representations, such as the design of heterogeneous and lightweight graph structures (Fan et al., 2025; Xu et al., 2025), the extension to hypergraphs that capturing higher-order relational dependencies (Luo et al., 2025; Feng et al., 2025), and the leverage of causal graphs to improve logical continuity (Wang et al., 2025). Additionally, retrieval strategies on graph KBs increasingly rely on heuristic path exploration, such as the topology-enhanced lightweight search (Fan et al., 2025), the pruning via relational path retrieval (Chen et al., 2025), the utilization of personalized memory-inspired reasoning (Jimenez Gutierrez et al., 2024; Gutiérrez et al., 2025), and the adoption of beam search over proposition paths (Wang, 2025). Despite these advances, current GraphRAG approaches remain constrained by shallow retrieval, limiting their ability to perform deep searching over graph KBs.

### 2.2 AGENTIC RETRIEVAL-AUGMENTED GENERATION

RAG improves factual grounding by retrieving external knowledge (Lewis et al., 2020), but single-round interaction is insufficient for complex reasoning tasks. Early advances focus on atomic-level improvements of RAG in query decomposition (Cao et al., 2023), query rewriting (Ma et al., 2023; Chan et al., 2024), retrieval compression (Xu et al., 2023), and selective retrieval decisions (Tan et al., 2024), which refine the retrieval process at a fine granularity. Beyond these, modular RAG systems (Gao et al., 2024; Jin et al., 2025b; Wu et al., 2025) have been proposed to flexibly reconfigure retrieval and reasoning modules into composable pipelines. More recently, agentic approaches emerged, enabling LLMs to iteratively plan, retrieve, and reflect. Representative methods include reasoning–acting synergy in ReAct (Yao et al., 2023), self-reflective retrieval in Self-RAG (Asai et al., 2024), test-time planning in PlanRAG (Verma et al., 2024), and reinforcement-learned search agents in Search-o1 (Li et al., 2025) and Search-r1 (Jin et al., 2025a). Subsequently, pioneering works (Sun et al., 2023; Ma et al., 2024; Shen et al., 2024; Lee et al., 2024) integrated structural graph knowledge for retrieval into the agentic RAG workflow to support the multi-step reasoning.

## 3 PRELIMINARIES

**Graph Knowledge Database.** Given a document collection $D$, the graph indexer $\phi$ segments $D$ into a set of text chunks $K$. For each chunk $k \in K$, an extractor $\mathcal{R} \in \phi$ identifies a set of entities $e = \{e_{\text{name}}, e_{\text{prop}}, e_{\text{desc}}\}$. For any pair of entities $e_h, e_t \in k$, a relation is defined as $r = \{e_h, e_t, r_{\text{prop}}, r_{\text{desc}}\}$. Aggregating all entities and relations yields the graph KB $G = \{E, R, K\}$, where $E$ denotes the entity set, $R$ the relation set, and $K$ the associated chunk-level textual context.

**Graph KB Retrieval.** Given a query $q$, a graph KB retriever $\psi$ selects a relevant context set $C = \{E_q, R_q, K_q\} \subset G$ that maximizes semantic relevance to $q$. The retriever aims to return structural graph data and chunk-based text data that provide sufficient evidence for answer generation.

**LLM Answer Generation.** The language model consumes the query $q$ together with the retrieved context $C$ to generate an output $y$. The generation is modeled as $P(y \mid q) = \sum_{C \subset G} P(y \mid q, C) \, P(C \mid q, G)$, where $P(C \mid q, G)$ represents the retrieval probability over the graph KB, and $P(y \mid q, C)$ denotes the generation probability conditioned on the integrated evidence.

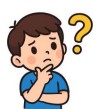

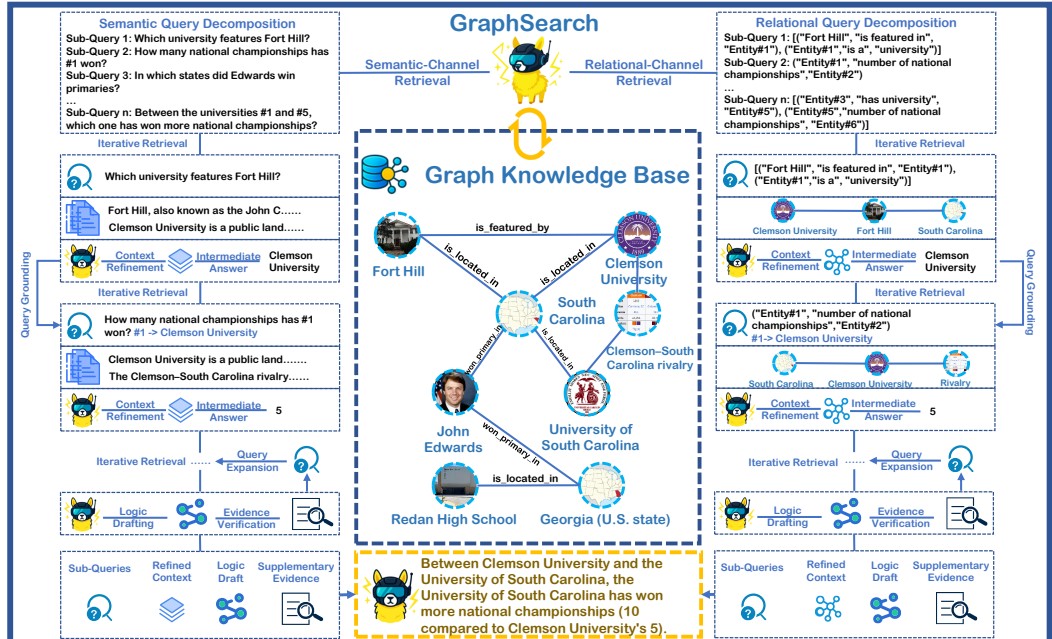

Figure 3: Overview of our GRAPHSEARCH framework.

# 4    GRAPHSEARCH

The overview of GRAPHSEARCH is shown in Figure 3. We build upon existing GraphRAG methods to construct the graph KB from documents. On top of this, GRAPHSEARCH leverages the GraphRAG retriever to perform deep searching, thereby enabling better answer generation.

## 4.1    THE MODULAR DEEP SEARCHING PIPELINE

### 4.1.1    ITERATIVE RETRIEVAL

**Query Decomposition.**    Given a complex query $Q$ as input, the goal of this module is to decompose $Q$ into a sequence of atomic sub-queries $\{q_1, q_2, \ldots, q_m\} = \text{P}_{\text{QD}}(Q)$ prompted by template $\text{P}_{\text{QD}}$, each representing a smaller and tractable component of the original question. In practice, each $q_i$ focuses on resolving a single entity, relation, or contextual dependency, thereby enabling the retriever to access fine-grained evidence and reducing the reasoning complexity of the overall task. For each sub-query $q_i$, the graph KB retriever $\psi$ accesses database $G$ to return

$$C_{q_i} = \psi(q_i \mid G) = \{E_{q_i}, R_{q_i}, K_{q_i}\} \tag{1}$$

where $C_{q_i}$ is the retrieved context of sub-query $q_i$. The detail of prompt $\text{P}_{\text{QD}}$ is in Figure 9.

**Context Refinement.**    Once the initial context $C_{q_i}$ is retrieved for a sub-query $q_i$, this module aims to refine the evidence by filtering redundant information and highlighting the most relevant entities, relations, and textual chunks. Given that raw retrieval, the refined context is obtained as $C'_{q_i} = \text{P}_{\text{CR}}(q_i, C_{q_i})$. This operation ensures that each refined context $C'_{q_i}$ contains only the most informative evidence for answering, thereby improving grounding quality in subsequent reasoning.

**Query Grounding.**    The sub-queries $\{q_1, q_2, \ldots, q_m\}$ are designed to be semantically independent yet logically ordered, such that the answer to one sub-query can serve as contextual grounding for subsequent ones. In practice, many decomposed queries may contain placeholders or unresolved

references that depend on the answers of prior sub-queries. To resolve this, each $q_i$ is first paired with its retrieved context $C_{q_i}$ and produce an intermediate answer $\hat{a}_{q_i} = \text{LLM}(q_i, C_{q_i})$, then progressively accumulated to support later queries. Formally, the grounded query is expressed as

$$\tilde{q}_i = \text{P}_{\text{QG}}(q_i, \{q_{<i}, C_{q_{<i}}, \hat{a}_{q_{<i}}\}), \tag{2}$$

This procedure guarantees that each $\tilde{q}_i$ is contextually instantiated rather than under-specified, enabling the graph KB retriever to fetch a more relevant context $C_{\tilde{q}_i}$ for subsequent reasoning.

### 4.1.2 Reflection Routing

**Logic Drafting.** The role of this module is to organize these pieces into a coherent reasoning chain that outlines how partial answers connect to the original query $Q$. Specifically, the drafting prompt $\text{P}_{\text{LD}}$ integrates the sequence of $\{q_i, C_{\tilde{q}_i}, \hat{a}_{q_i}\}$ to produce a structured draft $\mathcal{L}$, where

$$\mathcal{L} = \text{P}_{\text{LD}}(\{\tilde{q}_i, C_{\tilde{q}_i}, \hat{a}_{q_i}\}_{i=1}^m). \tag{3}$$

During this drafting process, the module not only consolidates available evidence but also exposes potential gaps in the reasoning chain. For instance, if a sub-query relies on entities or relations that were not retrieved in earlier steps, or if the accumulated sub-queries with intermediate answers $\{\tilde{q}_i, \hat{a}_{q_i}\}$ form an inconsistent chain, such deficiencies are explicitly reflected in $\mathcal{L}$ and exposed.

**Evidence Verification.** This module evaluates whether the accumulated evidence in $\mathcal{L}$ is sufficient and logically consistent to support a final answer. The verification prompt $\text{P}_{\text{EV}}$ inspects both the retrieved contexts and the intermediate answers, checking for factual grounding, coherence, and potential contradictions. Formally, this process can be described as

$$\mathcal{V} = \text{P}_{\text{EV}}(\{\tilde{q}_i, C_{\tilde{q}_i}, \hat{a}_{q_i}\}_{i=1}^m, \mathcal{L}), \tag{4}$$

where $\mathcal{V} \in \{\text{Accept}, \text{Reject}\}$ denotes the verification decision, the former implying that the reasoning chain is logically reliable, and the latter indicating missing or inconsistent evidence.

**Query Expansion.** This module generates additional sub-queries that explicitly target the missing evidence. Formally, using the expansion prompt and outputs a set of expanded sub-queries

$$\{q_j^+\}_{j=1}^n = \text{P}_{\text{QE}}(\{\tilde{q}_i, C_{\tilde{q}_i}, \hat{a}_{q_i}\}_{i=1}^m, \mathcal{L}). \tag{5}$$

Each expanded sub-query $q_j^+$ is submitted to the retriever $\psi$, yielding supplementary evidence $C_{q_i^+} = \psi(q_i^+ \mid G) = \{E_{q_i^+}, R_{q_i^+}, K_{q_i^+}\}$. The additional contexts $C_{q_i^+}$ are appended, thereby enriching the evidence pool and ensuring that knowledge gaps revealed in $\mathcal{L}$ can be actively filled, leading to a more reliable reasoning process.

### 4.2 Dual-Channel Retrieval

**Semantic Queries.** The semantic channel emphasizes retrieving descriptive evidence from chunk-level text. Given a complex query such as *"How many times did plague occur in the place where the creator of The Worship of Venus died?"*, the retriever first reformulates it into a sequence of semantically coherent sub-queries $\{q_1^{(s)}, q_2^{(s)}, \ldots, q_m^{(s)}\}$. Each $q_i^{(s)}$ is resolved against the text corpus as $C_{q_i^{(s)}} = \{K_{q_i^{(s)}}\}$, focusing on a single factual aspect, such as identifying the creator of the artwork, locating the place where this creator died, and finally retrieving records about the frequency of plague occurrences in that place. This design allows the semantic channel to capture nuanced descriptive information scattered across the corpus, ensuring that the retrieved textual evidence provides sufficient coverage for each step of reasoning.

**Relational Queries.** The relational channel formulates the same problem directly in terms of structured triples. Given a complex query $Q$, it is decomposed into a sequence of relational sub-queries $\{q_1^{(r)}, q_2^{(r)}, \ldots, q_n^{(r)}\}$, each mapped into subject–predicate–object relations. For each $q_j^{(r)}$, the retriever returns a subgraph context $C_{q_j^{(r)}} = \{E_{q_j^{(r)}}, R_{q_j^{(r)}}\}$, focusing only on entities and relations. For example, the painting *The Worship of Venus* $\rightarrow$ its creator $\rightarrow$ place of death $\rightarrow$ plague occurrences. Unresolved references (e.g., $\text{Entity}\#1, \text{Entity}\#2$) are progressively instantiated once upstream triples are resolved. This explicit traversal enforces logical dependencies and supports multi-hop reasoning, enabling the retriever to surface subgraphs that directly encode the answer path with reduced reliance on textual co-occurrence.

Table 1: Experiment results across six multi-hop QA benchmarks covering **Wikipedia**-based and **Domain**-based datasets. The + means **GRAPHSEARCH** integrates with various graph KB retrievers built upon the corresponding GraphRAG methods. The backbone LLM is *Qwen2.5-32B-Instruct*.

| Method | HotpotQA | | | MuSiQue | | | 2WikiMultiHopQA | | |
|---|---|---|---|---|---|---|---|---|---|
| | SubEM | A-Score | E-Score | SubEM | A-Score | E-Score | SubEM | A-Score | E-Score |
| Vanilla LLM | 33.67 | 6.90 | 5.98 | 12.33 | 6.10 | 5.87 | 48.33 | 6.95 | 4.50 |
| Naive RAG | 72.00 | 8.88 | 9.04 | 40.00 | 7.21 | 8.18 | 72.33 | 7.93 | 8.03 |
| ReAct | 33.33 | - | - | 16.00 | - | - | 51.33 | - | - |
| **GraphRAG Baselines** | | | | | | | | | |
| GraphRAG | 72.67 | 8.18 | 8.65 | 36.67 | 6.58 | 7.32 | 79.33 | 7.44 | 7.99 |
| LightRAG | 73.00 | 8.30 | 8.66 | 35.00 | 6.50 | 7.28 | 81.67 | 7.62 | 7.94 |
| MiniRAG | 68.00 | 7.95 | 8.24 | 41.00 | 6.93 | 7.67 | 74.00 | 7.57 | 7.61 |
| PathRAG | 79.00 | 8.99 | 9.17 | 46.33 | 7.26 | 8.02 | 77.00 | 8.25 | 8.34 |
| HippoRAG2 | 76.67 | 8.45 | 8.73 | 44.00 | 7.07 | 7.88 | 72.33 | 7.98 | 8.01 |
| HyperGraphRAG | 74.33 | 7.39 | 8.69 | 41.67 | 6.76 | 7.53 | 64.00 | 7.62 | 7.80 |
| **GRAPHSEARCH** | | | | | | | | | |
| + LightRAG | 79.00 | 9.21 | **9.46** | 51.00 | 7.72 | 8.38 | 85.00 | 9.21 | 9.12 |
| + PathRAG | **82.00** | **9.24** | 9.42 | **55.33** | **7.83** | **8.48** | **88.67** | **9.32** | **9.29** |
| + HyperGraphRAG | 80.33 | 9.19 | 9.35 | 49.33 | 7.73 | 8.22 | 83.33 | 8.84 | 8.75 |

| Method | Medicine | | | Agriculture | | | Legal | | |
|---|---|---|---|---|---|---|---|---|---|
| | SubEM | A-Score | E-Score | SubEM | A-Score | E-Score | SubEM | A-Score | E-Score |
| Vanilla LLM | 21.29 | 7.14 | 7.57 | 29.88 | 7.10 | 7.38 | 37.11 | 7.02 | 7.43 |
| Naive RAG | 54.34 | 8.23 | 8.67 | 54.24 | 7.91 | 8.26 | 53.36 | 7.37 | 7.67 |
| ReAct | 19.73 | - | - | 25.99 | - | - | 30.86 | - | - |
| **GraphRAG Baselines** | | | | | | | | | |
| GraphRAG | 53.32 | 7.59 | 7.98 | 57.81 | 7.84 | 7.66 | 58.98 | 7.57 | 7.23 |
| LightRAG | 49.80 | 7.36 | 7.57 | 55.66 | 7.38 | 7.32 | 56.84 | 7.01 | 6.78 |
| MiniRAG | 56.84 | 8.13 | 8.51 | 59.38 | 8.08 | 8.08 | 61.91 | 7.70 | 7.50 |
| PathRAG | 58.79 | 8.18 | 8.32 | 61.13 | 8.22 | 8.23 | 62.30 | 7.96 | 7.91 |
| HippoRAG2 | 55.08 | 7.90 | 8.03 | 58.20 | 7.95 | 7.86 | 64.45 | 8.02 | 7.81 |
| HyperGraphRAG | 62.11 | 8.39 | 8.70 | 63.67 | 8.35 | 8.49 | 66.60 | 8.18 | 8.18 |
| **GRAPHSEARCH** | | | | | | | | | |
| + LightRAG | 65.88 | 8.61 | 8.80 | 63.53 | 8.52 | 8.48 | 71.68 | 8.45 | 8.52 |
| + PathRAG | 70.12 | 8.59 | 8.82 | 69.34 | 8.63 | 8.78 | 74.41 | 8.32 | 8.49 |
| + HyperGraphRAG | **73.24** | **8.87** | **9.24** | **73.83** | **8.93** | **9.02** | **78.52** | **8.76** | **8.83** |

## 5 EXPERIMENTS

### 5.1 EXPERIMENTAL SETUP

**Datasets.** To evaluate the performance of GRAPHSEARCH, we conducted experiments on six multi-hop QA benchmarks within the RAG setting. The **Wikipedia**-based benchmarks include **HotpotQA** (Yang et al., 2018), **MuSiQue** (Trivedi et al., 2022), and **2WikiMultiHopQA** (Ho et al., 2020) following (Gutiérrez et al., 2025; Yang et al., 2025). The **Domain**-based benchmarks (Qian et al., 2025) incorporate multi-hop questions synthesized by (Luo et al., 2025), covering fields like **Medical**, **Agriculture**, and **Legal**. More details are provided in the Appendix C.

**Baselines.** We compare GRAPHSEARCH with several baseline methods, including **Vanilla LLM**, **Naive RAG** (Lewis et al., 2020), **GraphRAG** (Edge et al., 2024), **LightRAG** (Guo et al., 2024), **MiniRAG** (Fan et al., 2025), **PathRAG** Chen et al. (2025), **HippoRAG2** (Gutiérrez et al., 2025), and **HyperGraphRAG** (Luo et al., 2025). More details are provided in the Appendix D.

**Evaluation Metrics.** We adopt three evaluation metrics to assess the QA and retrieval quality of GRAPHSEARCH and baselines. The string-based Substring Exact-Match (**SubEM**) metric checks whether the golden answer is explicitly contained in the response. The Answer-Score (**A-Score**) covers **Correctness**, **Logical Coherence**, and **Comprehensiveness**. The Evidence-Score (**E-Score**) measures **Relevance**, **Knowledgeability**, and **Factuality**. Both A-Score and E-Score are assessed using the LLM-as-a-Judge (Gu et al., 2024). More details are provided in the Appendix F.

## 5.2 MAIN RESULTS

**GRAPHSEARCH outperforms all GraphRAG baselines.** As shown in Table 1, comparing with GraphRAG methods that perform only a single round of graph retrieval and generation, GRAPHSEARCH leverages the constructed graph knowledge bases with the graph KB retriever to enable multi-turn interactions. Across six benchmarks covering Wikipedia and domain-based datasets, **GRAPHSEARCH achieves the best overall performance**. This confirms the importance of adopting an agentic workflow for deep searching over GraphRAG in complex reasoning scenarios, effectively mitigating the insufficiencies of vanilla strategies caused by limited interaction and inadequate retrieval. Case studies with more detail information of are provided in Figure 11 and Figure 12 in Appendix B.

**GRAPHSEARCH exhibits strong plug-and-play capability.** As shown in Table 1, when applied with various retrieval methods over different graph KBs, GRAPHSEARCH consistently yields improvements compared to their native interaction schemes. For example, it boosts LightRAG on MuSiQue, raising SubEM from **35.00** to **51.00**, while improving A-Score and E-Score from **6.50** and **7.28** to **7.72** and **8.38**. Similarly, it enhances HyperGraphRAG on Medicine, increasing SubEM from **62.11** to **73.24**, and further elevating A-Score and E-Score from **8.39** and **8.70** to **8.87** and **9.24**. These results demonstrate the plug-and-play capability of GRAPHSEARCH, with detailed results presented in Figure 4.

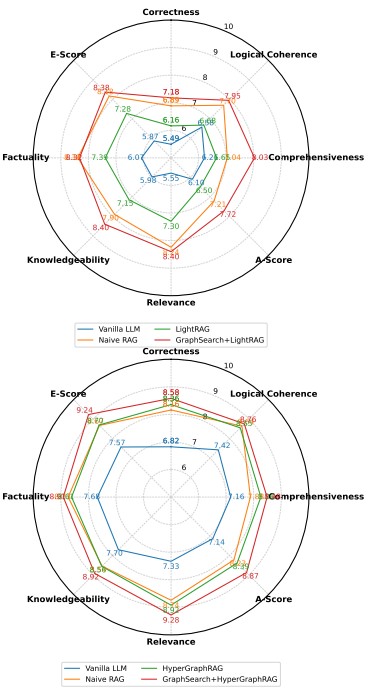

Figure 4: Judge results across eight metrics on A-Score and E-Score.

## 5.3 ABLATION STUDIES

**GRAPHSEARCH still remains effective under reduced model size.** Using *Qwen2.5-7B-Instruct* as the backbone, the experimental results on the **2Wiki.** and **Legal** datasets are reported in Table 2. Compared to three GraphRAG baselines, GRAPHSEARCH built upon these graph KB retrievers consistently achieves performance improvements. This confirms the potential of GRAPHSEARCH to extend effectively to models with reduced size.

**GRAPHSEARCH benefits from the design of dual-channel retrieval.** As shown in Figure 5, the QA performance on the **2Wiki** and **Legal** datasets obtained by integrating retrieval contexts from both channels consistently sur-

Table 2: Results across two benchmarks. The backbone LLM is *Qwen2.5-7B-Instruct*.

| Method | 2Wiki. | | | Legal | | |
|---|---|---|---|---|---|---|
| | SubEM | A-S | R-S | SubEM | A-S | R-S |
| Vanilla LLM | 46.67 | 6.26 | 3.70 | 34.18 | 6.47 | 6.89 |
| Naive RAG | 62.33 | 7.37 | 7.41 | 52.58 | 6.71 | 7.29 |
| **GraphRAG Baselines** | | | | | | |
| LightRAG | 72.33 | 7.11 | 7.53 | 52.93 | 6.50 | 6.45 |
| PathRAG | 73.00 | 7.44 | 7.71 | 58.98 | 7.06 | 7.01 |
| HyperGraphRAG | 72.33 | 7.49 | 7.69 | 60.11 | 7.32 | 7.19 |
| **GRAPHSEARCH** | | | | | | |
| + LightRAG | 79.00 | 8.35 | 8.21 | 58.59 | 7.64 | 7.31 |
| + PathRAG | 82.00 | **8.51** | 8.59 | 64.32 | 7.87 | 7.66 |
| + HyperGraphRAG | **82.33** | 8.49 | **8.69** | 67.48 | **8.02** | **7.39** |

passes that of either single-channel variant across all graph KB retrievers. The relative improvements between dual-channel retrieval and single-channel retrieval are particularly pronounced on the **Legal** dataset. This confirms **the necessity of the design of dual-channel retrieval**, which fully leverages the graph KBs constructed by GraphRAG from both semantic and structural perspectives.

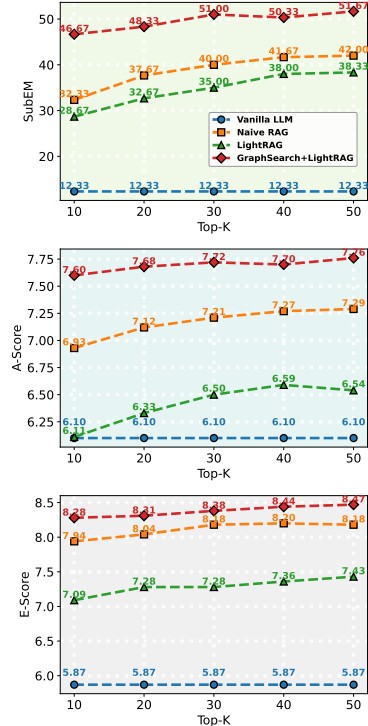

Figure 5: Comparisons between dual-channel and single-channel retrieval in GRAPHSEARCH, integrated with the graph KB retrievers built upon LightRAG, PathRAG and HyperGraphRAG.

Table 3: Experiment results of ablation study across 2Wiki. and Legal datasets of GRAPHSEARCH + HyperGraphRAG. ✓and / refer to whether each individual module is enable or not.

| Modules | | | | | | 2Wiki. | | | Legal | | |
|---|---|---|---|---|---|---|---|---|---|---|---|
| QD | CR | QG | LD | EV | QE | SubEM | A-Score | R-Score | SubEM | A-Score | R-Score |
| GRAPHSEARCH + HyperGraphRAG | | | | | | | | | | | |
| / | / | / | / | / | / | 64.00 | 7.62 | 7.80 | 66.60 | 8.18 | 8.18 |
| ✓ | ✓ | / | / | / | / | 76.33 | 8.14 | 8.16 | 73.98 | 8.34 | 8.29 |
| ✓ | ✓ | ✓ | / | / | / | 81.67 | 8.57 | 8.57 | 77.31 | **8.82** | 8.71 |
| ✓ | ✓ | ✓ | ✓ | / | / | 81.33 | 8.66 | **8.75** | 76.96 | 8.62 | 8.70 |
| ✓ | ✓ | ✓ | ✓ | ✓ | ✓ | **83.33** | **8.84** | **8.75** | **78.52** | 8.76 | **8.83** |

**GRAPHSEARCH modules make clear contributions to the agentic deep searching workflow.** We empirically evaluate the incremental contributions of the individual components in GRAPHSEARCH, including *Query Decomposition (QD)*, *Context Refinement (CR)*, *Query Grounding (QG)*, *Logic Drafting (LD)*, *Evidence Verification (EV)*, and *Query Expansion (QE)*. The design details of each module are provided in Appendix A. We adopt the graph KB retriever built upon HyperGraphRAG for GRAPHSEARCH along with HyperGraphRAG as a baseline. Comparing the combination of [QD, CR] with [QD, CR, QG], the former performs non-iterative question decomposition, producing multiple sub-queries without completing missing information based on retrieved context. Comparing [QD, CR, QG, AD] with the full-module setting, the former only introduces an additional logic drafting, whereas the latter further leverages reflection to generate new sub-queries that fill knowledge gaps. The empirical results confirm the value of the modular orchestration in GRAPHSEARCH: from question decomposition, to iterative retrieval, to reflective routing, each step progressively enhances the reasoning process and enables the realization of an agentic deep searching workflow.

**GRAPHSEARCH exhibits more pronounced advantages under smaller retrieval budgets.** By varying the **Top-K** from 10 to 50 as a adjustment strategy for retrieval overhead, the comparison of GRAPHSEARCH with baselines on MuSiQue is shown in Figure 6. As Top-K decreases, both Naive RAG and LightRAG show a sharp decline in SubEM and A-Score. In contrast, the drop in E-Score is less pronounced across all three methods, indicating that their retrievers can still capture part of the golden evidence under reduced budgets. However, the absence of critical evidence can prevent models from engaging in sufficient evidence-grounded reasoning, resulting in lower A-Scores relative to the golden answer. By contrast, the agentic searching workflow in **GRAPHSEARCH sustains its performance advantages even under low retrieval overhead**.

Figure 6: Performance changes as the count of Top-K varies.

### 5.4 FURTHER ANALYSIS: DEEP INTEGRATION OF GRAPHSEARCH WITH GRAPH KBS

**GRAPHSEARCH improves the retrieval quality through the dual-channel agentic interaction across both modalities.** Using **Recall** to calculate the golden evidence in the retrieved context, we compare the retrieval quality of GRAPHSEARCH with LightRAG, as shown in Figure 7. The **Step** denotes the interaction rounds performed by GRAPHSEARCH, up to the final self-reflection stage. GRAPHSEARCH initially retrieves fewer pieces of golden evidence, as it decomposes complex queries into atomic sub-queries. As interactions proceed, the recall of retrieved content shows substantial improvement across both the relational and semantic channels. It confirms that the agentic workflow of GRAPHSEARCH is tightly integrated with the features of graph KBs.

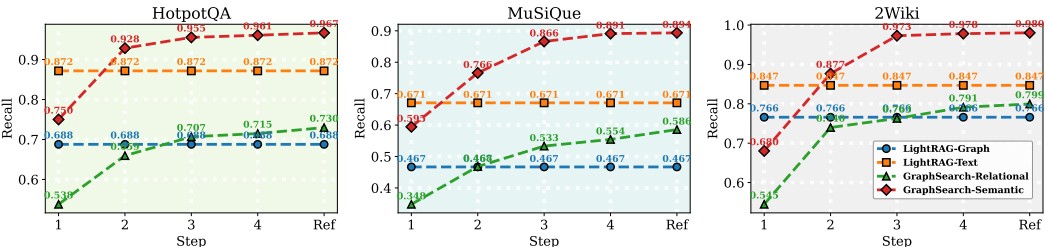

Figure 7: GRAPHSEARCH improves the recall of golden evidence during agentic interactions.

**GRAPHSEARCH demonstrates a modality–functionality alignment property in dual-channel retrieval.** We calculate SubEM on MuSiQue by replacing the retrieval sources of the semantic and relational channel with text and graph data respectively. Results obtained by retrieving from the full data are included as references. Figure 8 shows that using semantic queries to access text data and relational queries to access graph data consistently outperforms other combinations. Moreover, compared to retrieving from the full data, restricting each channel to its aligned modality not only achieves comparable performance but also substantially reduces context overhead. It confirms that the functionality of the dual-channel retrieval strategy aligns with the data modalities of graph KBs.

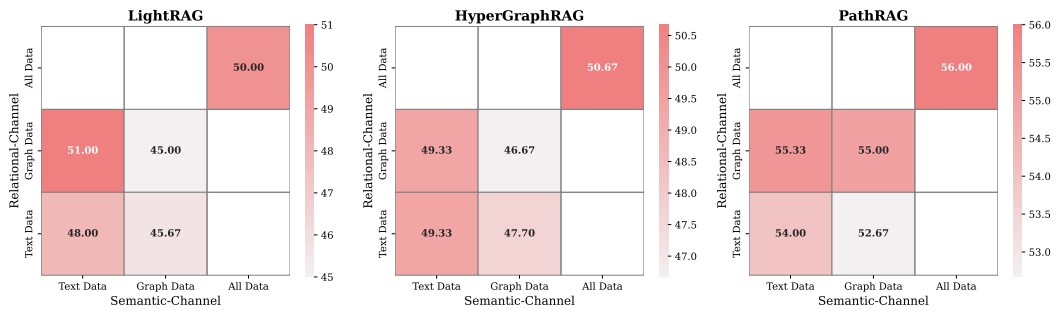

Figure 8: GRAPHSEARCH demonstrates a modality–function alignment property.

## 6 CONCLUSION

In this work, we introduced GRAPHSEARCH, a novel agentic deep searching framework for GraphRAG. By integrating dual-channel retrieval over both semantic text chunks and structural graph data, GRAPHSEARCH effectively overcomes the limitations of shallow retrieval and inefficient graph utilization. Its modular design enables iterative reasoning and multi-turn interactions, leading to more comprehensive evidence aggregation. Experimental results on six multi-hop RAG benchmarks demonstrate consistent improvements in answer accuracy and generation quality, highlighting the effectiveness of our approach. We believe GRAPHSEARCH offers a promising direction for advancing graph retrieval-augmented generation.

## ETHICS STATEMENT

Our work builds upon publicly available text corpora for constructing graph-based indices in the context of retrieval-augmented generation. While we have taken care to rely on community-curated and open datasets, it is possible that a small portion of the data may contain biases, fairness issues, or inadvertent privacy leaks. Moreover, once GRAPHSEARCH is released as open source, we cannot fully prevent the community from applying it to corpora that may raise ethical concerns, such as those containing sensitive or non-consensual information. To mitigate such risks, we will provide clear documentation and guidelines for responsible use, encourage the community to exercise caution in dataset selection, and call for future research on automated ethical auditing methods to ensure fairness, privacy protection, and compliance in knowledge graph-based retrieval systems.

## REPRODUCIBILITY STATEMENT

We have provided detailed descriptions of our implementation in the Implementation Details section, including preprocessing procedures, dataset and model selection, experimental hyperparameters, and the executing environment. Due to the large scale of the datasets and the need to preserve anonymity during the double-blind review process, we do not release code or processed datasets at this stage. However, we commit to releasing executable code and the processed datasets after the review process is completed, ensuring full reproducibility of our results.

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

# APPENDIX

## A    PROMPT TEMPLATES

As shown in Figure 9, we introduce the prompt templates in **Query Decomposition**, **Context Refinement** and **Query Rewriting** modules both in text-channel and graph-channel.

---

**Text-Channel Query Decomposition**

```
"""---Role---

You are a helpful assistant specializing in complex query decomposition.

---Goal---

Given a main query, your task is to break it down into several atomic sub-queries, which should
directly correspond to parts of the original query.

---Instructions---

- Decompose the main query into clear and actionable sub-queries that represent smaller, solvable
pieces of the main question.
- Ensure that each sub-query addresses one specific entity or concept, with the goal of retrieving
information that will answer the overall main query.
- Use sequential numbering (i.e., `#1`, `#2`, etc.) to represent answers of previous sub-queries. For
example, `#1` refers to the answer of Sub-query 1.
- Make sure the sub-queries are logically ordered, where the output of one sub-query might feed into
the next.
- The final output should be in JSON format, where each sub-query is listed as a key-value pair.

---Examples---

Main Query:
How many times did plague occur in the place where the creator of The Worship of Venus died?

Sub-queries:
{{
    "Sub-query 1": "Who is the creator of The Worship of Venus?",
    "Sub-query 2": "Where did #1 die?",
    "Sub-query 3": "How many times did plague occur in #2?"
}}

Main Query:
When did the city where Hillcrest High School is located become the capital of the state where the
screenwriter of The Poor Boob was born?

Sub-queries:
{{
    "Sub-query 1": "Where is Hillcrest High School located?",
    "Sub-query 2": "Who is the screenwriter of The Poor Boob?",
    "Sub-query 3": "Where was #2 born?",
    "Sub-query 4": "When did the city from #1 become the capital of the state from #3?"
}}

Main Query:
What crop, which is a big feeder of nitrogen, has a gross income of $1,363.00 per acre and a net profit
of $658.00?

Sub-queries:
{{
    "Sub-query 1": "Which crops are considered big feeders of nitrogen?",
    "Sub-query 2": "Among #1, which crop has a gross income of $1,363.00 per acre?",
    "Sub-query 3": "Does #2 have a net profit of $658.00?"
}}

---Input---

Main Query:
{query}

---Output---
"""
```

**Graph-Channel Query Decomposition**

```
"""---Role---

You are a helpful assistant specializing in complex query decomposition for knowledge graph retrieval.

---Goal---

Given a main query, your task is to break it down into atomic sub-queries in the form of subject-
predicate-object triples. These should correspond directly to parts of the original query and be suitable
for querying a knowledge graph.

---Instructions---

- Decompose the main query into a sequence of sub-queries, where each sub-query consists of one or more
atomic triples in the format: ("entity1", "relationship", "entity2").
- Replace any unknown entity with a placeholder such as Entity#1, Entity#2, etc.
- Maintain logical ordering, where the result of one sub-query (e.g., Entity#1) might be required for the
next.
- Each sub-query may contain more than one triple if needed to express the full meaning.
- The final output should be in JSON format, where each key is a sub-query and the value is a list of
atomic triples enclosed in parentheses.

---Examples---

Main Query:
How many times did plague occur in the place where the creator of The Worship of Venus died?

Sub-queries:
{{
    "Sub-query 1": [("The Worship of Venus", "is created by", "Entity#1")],
    "Sub-query 2": [("Entity#1", "died at", "Entity#2")],
    "Sub-query 3": [
        ("Plague", "occur in", "Entity#2"),
        ("Plague", "times of occur", "Entity#3")
    ]
}}

Main Query:
When did the city where Hillcrest High School is located become the capital of the state where the
screenwriter of The Poor Boob was born?

Sub-queries:
{{
    "Sub-query 1": [("Hillcrest High School", "is located in", "Entity#1")],
    "Sub-query 2": [("The Poor Boob", "has screenwriter", "Entity#2")],
    "Sub-query 3": [("Entity#2", "was born in", "Entity#3")],
    "Sub-query 4": [
        ("Entity#1", "is capital of", "Entity#3"),
        ("Entity#1", "became capital at", "Entity#4")
    ]
}}

Main Query:
What crop, which is a big feeder of nitrogen, has a gross income of $1,363.00 per acre and a net profit
of $658.00?

Sub-queries:
{{
    "Sub-query 1": [("Entity#1", "is a", "crop that is a heavy nitrogen feeder")],
    "Sub-query 2": [("Entity#1", "has gross income per acre", "$1,363.00")],
    "Sub-query 3": [("Entity#1", "has net profit", "$658.00")]
}}

---Input---

Main Query:
{query}

---Output---
"""
```

---

**Text-Channel Context Refinement**

```
"""---Role---

You are a helpful summarizer specialized in extracting relevant evidence from retrieved documents.

---Goal---

Given a user query and retrieved context, your task is to produce a comprehensive summary from context
data that highlights all potentially useful information relevant to answering the user query.

---Instructions---

- Carefully analyze the context data for facts, arguments, or examples that align with the query.
- Organize the output in a well-structured paragraph.
- Do not speculate or introduce information not found in the context.

---Input---

User-Query:
{query}

Context Data:
{context_data}

---Output---
"""
```

**Graph-Channel Context Refinement**

```
"""---Role---

You are a helpful knowledge graph extractor specialized in identifying relevant knowledge triplets from
retrieved graph data.

---Goal---

Given a user query and retrieved knowledge graph data, your task is to extract all relevant knowledge
triplets from graph data that highlights all potentially useful information relevant to answering the user
query.

---Instructions---

- Carefully examine the knowledge graph data to identify triplets (entity1, relationship, entity2)
directly related to the user query.
- Do not infer or generate information beyond the given data.
- Format the output strictly as a list of JSON triplets, each in the following form:
  [("entity1", "relationship", "entity2"), ...]

---Input---

User-Query:
{query}

Knowledge Graph Data:
{context_data}

---Output---
"""
```

---

**Text-Channel Query Grounding**

```
"""---Role---

You are a helpful assistant specializing in completing partially defined sub-queries using prior context.

---Goal---

Given a sub-query containing placeholders like #1, #2, etc., and the context of previous sub-queries with
retrieved results, your task is to replace the references (e.g., #1) with the actual answers from the
context.

Your output should be a fully resolved and standalone query. If the placeholder cannot be resolved with
the context, leave the sub-query unchanged.

---Input---

Sub-query:
{sub_query}

Context Data:
{context_data}

---Output---
"""
```

**Graph-Channel Query Grounding**

```
"""---Role---

You are a helpful assistant specializing in completing partially defined knowledge graph sub-queries using
prior context.

---Goal---

Given a sub-query containing placeholders like Entity#1, Entity#2, etc., and the context providing actual
values for these placeholders, your task is to replace the placeholders with the corresponding entities if
available.

Your output should maintain the same format as the original sub-query. If the placeholder cannot be
resolved with the context, leave the sub-query unchanged.

---Input---

Sub-query:
{sub_query}

Context Data:
{context_data}

---Output---
"""
```

Figure 9: Prompt templates of **Query Decomposition**, **Context Refinement** and **Query Rewriting** modules both in text-channel and graph-channel.

As shown in Figure 10, we introduce the prompt templates in **Logic Drafting**, **Evidence Verification** and **Query Expansion** modules for combining into a reflection router.

```
Logic Drafting

"""---Role---

You are a helpful assistant specializing in complex question answering.

---Goal---

Given a complex query and retrieved context data, your task is to construct a logically sound, step-by-step answer.
Your explanation should follow a rigorous reasoning path, incorporate relevant evidence, and establish clear relationships between the
entities.

---Instructions---

- Break down the reasoning process into clear, coherent steps.
- Use context data explicitly to support each reasoning step.
- Make sure relationships between entities are logically explained.

---Input---

Query:
{query}

Context Data:
{context_data}

---Output---
"""
```

```
Evidence Verification

"""---Role---

You are a critical evaluator specializing in verifying the logical soundness and evidential sufficiency of model-generated responses.

---Goal---

Given a user query, retrieved context data, and the model-generated response, your task is to evaluate whether the response forms a rigorous
logical loop supported by the provided evidence.

---Instructions---

- Carefully examine whether the response is **strictly grounded** in the retrieved context data.
- Assess whether the reasoning process forms a **complete logical chain**, without missing steps or unsupported leaps.
- Identify if there are **evidence gaps, low-confidence claims, or speculative statements**.
- If the response demonstrates a well-supported, confident, and logically closed argument, conclude your analysis with **"Yes"**.
- If the response shows hesitation, incomplete reasoning, or lacks solid evidence support, conclude your analysis with **"No"**.

---Input---

User-Query:
{query}

Retrieved Context Data:
{context_data}

Model Response:
{model_response}

---Output---
"""
```

```
Query Expansion

"""---Role---

You are a helpful assistant specializing in query expansion for evidence completion.

---Goal---

Given a main query, retrieved context data, the model-generated response, and the evidence verification analysis, your task is to perform
**query expansion**.
If the evidence verification analysis shows that the current evidence is insufficient to support the logical chain of the response, generate
one or more additional sub-queries.
These sub-queries should aim to cover missing retrieval scenarios, fill in the evidence gaps, and guide towards a more complete and
confident logical reasoning chain.

---Instructions---

- Use the retrieved context data, especially any existing sub-queries in the retrieval history, as references when generating new sub-
queries.
- Focus on producing **complementary sub-queries** that address aspects not yet fully supported by evidence.
- Avoid duplicating existing sub-queries; instead, expand into related but uncovered areas.
- Keep sub-queries clear, specific, and directly actionable for retrieval.
- Output should be in the form of a **Python-style List of strings**, where each string is a new sub-query.

---Input---

Main Query:
{query}

Retrieved Context Data:
{context_data}

Model Response:
{model_response}

Evidence Verification Analysis:
{evidence_verification}

---Output---
"""
```

Figure 10: Prompt templates of **Logic Drafting**, **Evidence Verification** and **Query Expansion** modules for combining into a reflection router.

# B    CASE STUDIES

As shown in Figure 11, there are some cases of baseline methods, including vanilla LLM generation, Naïve RAG and LightRAG. A case of our proposed GRAPHSEARCH is in Figure 12.

---

**Question: Who won more national championships between the university featuring Fort Hill and the university of the state where Edwards won the primary besides the state containing Redan High School?**

**Answer: University of South Carolina**

**Vanilla LLM**

To answer this question accurately, we need to break it down into parts and identify the specific universities and states being referred to.\n\n1. **University featuring Fort Hill**: Fort Hill is located in Baltimore, Maryland. The university most closely associated with Baltimore is Johns Hopkins University.\n2. **University of the state where Edwards won ……

Therefore, based on the information provided and the assumptions made, the university in **North Carolina** (likely referring to either Duke University or UNC) would have won more national championships compared to Johns Hopkins University.

**Naïve RAG**

| Context | | | | |
|---|---|---|---|---|
| Fort Hill High School: Fort Hill High School is a four-year public high school in Cumberland, Maryland, United States, serving 780 students in grades 9 through 12. | Redan High School: Redan High School is a public secondary school of the DeKalb County School District located in unincorporated DeKalb County, Georgia, United States. …… | Maynard Evans High School: Maynard Evans High School is a high school located in Orlando, Florida, United States, served by Orange County Public Schools. …… | Herbert Henry Dow High School: Herbert Henry Dow High School is a public high school located in Midland, Michigan. The school, a part of Midland Public Schools, …… | Fort Hill (Clemson, South Carolina): Fort Hill, also known as the John C. Calhoun Mansion and Library, is a National Historic Landmark on the Clemson University campus in Clemson…… |

**Answer**

The query asks about comparing the number of national championships between two universities: one featuring Fort Hill and another in the state where Edwards won a primary, excluding the state that contains Redan High School.\n\nFrom the provided context:\n- Fort Hill is associated with Clemson University in South Carolina.\n- Edwards won the South Carolina ……

**Since no specific information** about Clemson University's national championships is given, and the University of Kansas is the only university mentioned with a clear number of national championships, it can be inferred that the University of Kansas has won more national championships than Clemson University based on the available information.

**LightRAG**

**Retrieved Graph Data**

{\"id\": 1, \"entity\": \"DeKalb County School District\", \"type\": \"organization\", \"description\": \"DeKalb County School District is the educational authority overseeing Redan High School.\", \"rank\": 1, \"created_at\": \"2025-08-07 17:23:12\", \"file_path\": \"unknown_source\"}, {\"id\": 2, \"entity\": \"Redan High School\", \"type\": \"organization\", \"description\": \"Redan High School is a public secondary school within the DeKalb County School District.\", \"rank\": 3, \"created_at\": \"2025-08-07 17:23:12\", \"file_path\": \"unknown_source\"}, {\"id\": 3, \"entity\": \"Grades 9 through 12\", \"type\": \"category\", \"description\": \"This category represents the academic levels served by Fort Hill High School, ranging from freshmen to seniors.\", \"rank\": 1, \"created_at\": \"2025-08-07 07:53:00\", \"file_path\": \"unknown_source\"}……

{\"id\": 1, \"entity1\": \"Fort Hill High School\", \"entity2\": \"United States\", \"description\": \"Fort Hill High School is a public high school in the United States.\", \"keywords\": \"educational system,national context\", \"weight\": 7.0, \"rank\": 476, \"created_at\": \"2025-08-07 07:53:00\", \"file_path\": \"unknown_source\"}, {\"id\": 2, \"entity1\": \"Community High School District 100\", \"entity2\": \"United States\", \"description\": \"Community High School District 100 operates within the United States.\", \"keywords\": \"geographical containment,national jurisdiction\", \"weight\": 7.0, \"rank\": 473, \"created_at\": \"2025-08-07 11:35:55\", \"file_path\": \"unknown_source\"}, {\"id\": 3, \"entity1\": \"DeKalb County\", \"entity2\": \"Georgia\", \"description\": \"DeKalb County is located within the state of Georgia.\", \"keywords\": \"geographical containment,state affiliation\"……

**Retrieved Text Data**

{\"id\": 1, \"content\": \"Redan High School: Redan High School is a public secondary school of the DeKalb County School District located in unincorporated DeKalb County, Georgia, United States. The school is south of the city of Stone Mountain.\", \"file_path\": \"unknown_source\"}, {\"id\": 2, \"content\": \"Fort Hill High School: Fort Hill High School is a four-year public high school in Cumberland, Maryland, United States, serving 780 students in grades 9 through 12.\", \"file_path\": \"unknown_source\"}, {\"id\": 3, \"content\": \"William Howard Taft High School (Chicago): Taft High School is a public 6-year high school and academic center located in the Norwood Park neighborhood on the northwest side of Chicago, Illinois, United States. Taft serves communities on the far northwest side, specifically Norwood Park, Edison Park, Jefferson Park, Forest Glen and O'Hare. Taft is operated by the Chicago Public Schools district. The school is perhaps most famous as the high school attended by Jim Jacobs, the writer of \\\"Grease\\\". Jacobs used Taft as an inspiration in writing the musical. Taft's NJROTC unit has won a Distinguished Unit award every year since 2001. Since 2014, Taft High School has been considered a \\\"wall-to-wall\\\" IB school as part of the International Baccalaureate foundation. Taft high school has offered the International Baccalaureate program since 2001.\", \"file_path\": \"unknown_source\"}……

**Answer**

Based on the provided information, **there is no mention of any specific universities** associated with Fort Hill High School or any primary elections involving Edwards. The entities related to Fort Hill High School are primarily focused on its location and academic details in Cumberland, Maryland, while Redan High School is mentioned in the context of being part of the DeKalb County School District in Georgia. No information regarding national championships or political primaries involving Edwards is provided in the given Knowledge Graph and Document Chunks.\n\nTherefore, it is not possible to determine who won more national championships between these entities based on the available data.

Figure 11: Samples of Vanilla Generation, Naive RAG and LightRAG.

**Question: Who won more national championships between the university featuring Fort Hill and the university of the state where Edwards won the primary besides the state containing Redan High School?**

**Answer: University of South Carolina**

### GraphSearch + LightRAG

| Semantic Query Decomposition | Relational Query Decomposition |
|---|---|
| "Which university features Fort Hill?", "Who is Edwards?", "In which states did Edwards win primaries?", "Which state containing Redan High School should be excluded from the list obtained in #3?", "Identify the university in the remaining state(s) from #3 where Edwards won a primary.", "How many national championships has the university identified in #1 won?", "How many national championships has the university identified in #5 won?", "Between the universities from #1 and #5, which one has won more national championships?" | [("Fort Hill", "is featured in", "Entity#1"), ("Entity#1", "is a", "university")] [("Entity#1", "number of national championships", "Entity#2")] [("Edwards", "won primary in", "Entity#3"), ("Redan High School", "is located in", "Entity#4"), ("Entity#3", "is not", "Entity#4")] [("Entity#3", "has university", "Entity#5"), ("Entity#5", "number of national championships", "Entity#6")] [("Entity#2", "compare with", "Entity#6")] |

| Semantic-Channel Retrieval | Relational-Channel Retrieval |
|---|---|
| Which university features Fort Hill? | [("Fort Hill", "is featured in", "Entity#1"), ("Entity#1", "is a", "university")] |
| {\"id\": 1, \"content\": \"Fort Hill (Clemson, South Carolina): Fort Hill, also known as the John C. Calhoun Mansion and Library, is a National Historic Landmark on the Clemson University campus in Clemson, South Carolina. The house is significant as the home from 1825-50 of John C. Calhoun, a leading national politician of the period, and is now a museum and library maintained in his memory.\", \"file_path\": \"unknown_source\"}, {\"id\": 2, \"content\": \"Memorial Union (Oregon State University): The Memorial Union (MU) is the student activity center at Oregon State University in Corvallis, …… | {\"id\": 1, \"entity\": \"Historic Places\", \"type\": \"category\", \"description\": \"A category representing sites recognized for their historical significance.\", \"rank\": 1, \"created_at\": \"2025-08-07 06:51:31\", \"file_path\": \"unknown_source\"}, {\"id\": 2, \"entity\": \"Historic Properties\", \"type\": \"category\", \"description\": \"Historic Properties refer to buildings, structures, objects, sites, and districts recognized for their historical, architectural, and cultural significance.<SEP>Historic Properties refer to buildings, structures, …… |
| Fort Hill is featured on the Clemson University campus in Clemson, South Carolina. | [("Fort Hill", "is featured in", "Clemson University"), ("Clemson University", "is a", "university")] |
| Who is Edwards? | [("Clemson University", "number of national championships", "Entity#2")] |
| {\"id\": 1, \"content\": \"2004 United States presidential election: The following week, John Edwards won the South Carolina primary and finished a strong second in Oklahoma to Clark. Lieberman dropped out of the campaign the following day. Kerry dominated throughout February and his support quickly snowballed as he won caucuses and primaries, taking in a string of wins in Michigan, Washington, Maine, Tennessee, Washington, D.C., Nevada, Wisconsin, Utah, Hawaii, and Idaho. Clark and Dean dropped out during this time, leaving Edwards as the only real threat to Kerry. …… | {\"id\": 1, \"entity\": \"league title\", \"type\": \"category\", \"description\": \"A championship awarded to the top-performing team in the English football system.\", \"rank\": 0, \"created_at\": \"2025-08-07 02:50:16\", \"file_path\": \"unknown_source\"}, {\"id\": 2, \"entity\": \"champion\", \"type\": \"category\", \"description\": \"The champion is the highest-ranked club at the end of the season based on the total points accumulated.\", \"rank\": 1, \"created_at\": \"2025-08-07 02…… |
| Based on the provided context, there are two notable individuals named Edwards:\n\n1. **John Edwards**:…… | [("Clemson University", "number of national championships", "5 national championships")] |
| …… | …… |
| How many national championships has Clemson University won? | [("South Carolina", "has university", "Entity#5"), ("Entity#5", "number of national championships", "Entity#6")] |
| {\"id\": 1, \"content\": \"Clemson-South Carolina rivalry: College comparison Clemson University University of South Carolina Location Clemson Columbia Students 23,406 34,618 School colors Clemson Orange and Regalia Garnet and Black Nickname Tigers Gamecocks Mascot The Tiger and The Tiger Cub Cocky National Championships 5 10\", \"file_path\": \"unknown_source\"}, {\"id\": 2, \"content\": \"Fort Hill (Clemson, South Carolina): Fort Hill, also known as the John C. Calhoun Mansion and Library, is a National Historic Landmark on the Clemson University campus in Clemson, South Carolina. …… | {\"id\": 1, \"entity\": \"Redan High School\", \"type\": \"organization\", \"description\": \"Redan High School is a public secondary school within the DeKalb County School District.\", \"rank\": 3, \"created_at\": \"2025-08-07 17:23:12\", \"file_path\": \"unknown_source\"}, {\"id\": 2, \"entity\": \"DeKalb County School District\", \"type\": \"organization\", \"description\": \"DeKalb County School District is the educational authority overseeing Redan High School.\", \"rank\": 1, \"created_at\": \"2025-08-07 17:23:12\", \"file_path\": \"unknown_source\"}, {\"id\": 3, \"entity\": \"DeKalb County\", …… |
| Clemson University has won 5 national championships. | [("South Carolina", "has university", "University of South Carolina"), ("University of South Carolina", "number of national championships", "10 national championships")] |
| Between the universities from Clemson University and the University of South Carolina, which one has won more national championships? | [("5 national championships", "compare with", "10 national championships")] |
| {\"id\": 1, \"content\": \"Clemson-South Carolina rivalry: College comparison Clemson University University of South Carolina Location Clemson Columbia Students 23,406 34,618 School colors Clemson Orange and Regalia Garnet and Black Nickname Tigers Gamecocks Mascot The Tiger and The Tiger Cub Cocky National Championships 5 10\", \"file_path\": \"unknown_source\"}, {\"id\": 2, \"content\": \"Fort Hill (Clemson, South Carolina): Fort Hill, also known as the John C. Calhoun Mansion and Library, is a National Historic Landmark on the Clemson University campus…… | {\"id\": 1, \"entity\": \"Championship Titles\", \"type\": \"category\", \"description\": \"Championship Titles refer to the number of times a team or player has won the NBA championship.\", \"rank\": 2, \"created_at\": \"2025-08-07 12:38:38\", \"file_path\": \"unknown_source\"}, {\"id\": 2, \"entity\": \"league title\", \"type\": \"category\", \"description\": \"A championship awarded to the top-performing team in the English football league system.\", \"rank\": 0, \"created_at\": \"2025-08-07 02:50:16\", \"file_path\": \"unknown_source\"}, {\"id\": 3, \"entity\": \"champion\", \"type\":…… |
| Based on the provided context, the University of South Carolina has won more national championships than Clemson University. Specifically, Clemson University has won 5 national championships, whereas the University of South Carolina has won 10 national championships. | [("5 national championships", "compare with", "10 national championships")] |

| Logic Drafting and Evidence Verification | Logic Drafting and Evidence Verification |
|---|---|
| To address the query, we need to compare the number of national championships won by two universities: the one featuring Fort Hill and the university in the state where Edwards won a primary, excluding the state containing Redan High School.\n\n**Step 1: Identify the University Featuring Fort Hill**\n- From Sub-query 1, we know that Fort Hill is located on the Clemson University campus …… **Conclusion:**\nBetween Clemson University and the University of South Carolina, the University of South Carolina has won more national championships (10 compared to Clemson University's 5).", | To solve the query, we must compare the number of national championships between two universities: the one featuring Fort Hill and the one in the state where Edwards won a primary, excluding the state containing Redan High School.\n\nStep 1: University Featuring Fort Hill\n- From Sub-query 1, Fort Hill (Clemson, South Carolina) is located on the Clemson University campus. Therefore, Entity#1 = Clemson University. Since Clemson University is itself the …… Conclusion:\nThe University of South Carolina has won more national championships than Clemson University (10 vs. 5). |
| **Yes** | **Yes** |

| Answer |
|---|
| To address the query and provide a logically sound, step-by-step answer, we need to break down the problem into smaller parts and use the context data provided to find the necessary information.\n\n### Step 1: Identify the University Featuring Fort Hill\n- **Context Data**: Fort Hill is located on the Clemson University campus in Clemson, South Carolina.\n- **Conclusion**: The university featuring Fort Hill is Clemson University.\n\n### Step 2: Determine the State Where Edwards Won a Primary Election\n- …… **Conclusion**: The University of South Carolina has won more national championships than Clemson University.\n\n### Final Answer\nBetween the university featuring Fort Hill (Clemson University) and the university of the state where Edwards won the primary (University of South Carolina), the University of South Carolina has won more national championships.", |

Figure 12: A sample of GRAPHSEARCH with LightRAG as the graph retriever.

## C    DATASETS

As shown in Table C, we sample 300 questions for HotpotQA, MuSiQue and 2WikiMultiHopQA datasets, and directly adopt the Medicine, Agriculture and Legal datasets from (Luo et al., 2025).

Table 4: Detail information of datasets used in GRAPHSEARCH. The tokenizer used to calculate the size of corpora is GPT-4o. # means the number of counts.

| Name | Reference | Source | #Corpus | #Questions | Question Types | #Evidence |
|------|-----------|--------|---------|------------|----------------|-----------|
| HotpotQA | (Yang et al., 2018) | Wikipedia | 397,274 | 300 | Comparison, Bridge | 2,3,4 |
| MuSiQue | (Trivedi et al., 2022) | Wikipedia | 533,145 | 300 | 2-Hop, 3-Hop, 4-Hop | 2,4 |
| 2WikiMultiHopQA | (Ho et al., 2020) | Wikipedia | 220,295 | 300 | Compositional, Comparison, Bridge Comparison, Inference | 2,4 |
| Medicine | (McEvoy et al., 2024) | ESC Guidelines | 175,216 | 512 | 1-Hop, 2-Hop, 3-Hop | 1,2,3 |
| Agriculture | (Qian et al., 2025) | UltraDomain | 378,592 | 512 | 1-Hop, 2-Hop, 3-Hop | 1,2,3 |
| Legal | (Qian et al., 2025) | UltraDomain | 929,396 | 512 | 1-Hop, 2-Hop, 3-Hop | 1,2,3 |

## D    BASELINES

- **Vanilla LLM**: Zero-shot question and answering without any external retrieval source, depending on language model's parametric knowledge.
- **Naive RAG** (Lewis et al., 2020): Generation with plain text chunk-based embedding database as external retrieval source, where top-k items are retrieved for a single round.
- **GraphRAG** (Edge et al., 2024): A graph-based approach to question answering over hierarchical graph index where community summary is generated to represent the relationships.
- **LightRAG** (Guo et al., 2024): A simple and fast GraphRAG framework by applying integration of graph structures with vector representations for a dual-level retrieval system.
- **MiniRAG** (Fan et al., 2025): A novel GraphRAG system designed for small LLM which adopts a lightweight topology-enhanced retrieval approach.
- **PathRAG** (Chen et al., 2025): A GraphRAG system which retrieves key relational paths from the indexing graph through flow-based pruning.
- **HippoRAG2** (Gutiérrez et al., 2025): A RAG framework built upon the personalized PageRank with deeper passage integration.
- **HyperGraphRAG** (Luo et al., 2025): A novel hypergraph-based RAG method that represents n-ary relational facts via hyper-edges for retrieval and generation.

## E    IMPLEMENTATION DETAILS.

We conduct experiments on a Linux server equipped with 8 A100-SXM4-40GB GPUs. The model for graph construction is *Qwen2.5-32B-Instruct*, and the chunk size is 400 tokens. The embedding model for Naive-RAG and GraphRAG is *jinaai/jina-embeddings-v3* (Sturua et al., 2024). For GRAPHSEARCH and baselines, we set the **Hybrid** retrieval mode and set the **Top-K** for retrieval to 30, or use the default configuration if unavailable. The backbone model for generation is *Qwen2.5-7B/32B-Instruct* (Bai et al., 2023). The LLM-as-a-Judge for evaluation is *Qwen-Plus* (Bai et al., 2023), a strong closed-source model with API available.

## F    EVALUATION DETAILS

Inspired by (Yang et al., 2025; Luo et al., 2025), we leverage the Substring Extract-Match(**SubEM**) metric to check whether the golden answer is explicitly contained in the response, the Answer-Score(**A-Score**) to judge the quality of model generation across 3 criteria covering **correctness**, **logical coherence** and **comprehensiveness** with the **golden answer** as reference, and the Evidence-Score(**E-Score**) to measure how well the model's generation is grounded in the golden evidence, evaluated along 3 criteria including **relevance**, **knowledgeability** and **factuality** with the **golden evidence** as reference as follows:

$$\text{SubEM} = \frac{1}{N} \sum_{i=1}^{N} \mathbf{1}\left[\text{contains}\left(O_i^{\text{pred}}, A_i^{\text{gold}}\right)\right], \tag{6}$$

```
"""---Role---
You are a helpful and rigorous
assistant evaluating the
**{title}** of a generated response.
---Question---
{question}
---Golden Answer---
{gold_answer}
---Evaluation Goal---
Evaluate **{goal}** using a **0-10
integer scale**.
{rubric}
---Output Format---
Score (an integer from 0 to 10)
---Generation to be Evaluated---
{response}
"""
```

```
"correctness": (
        "correctness",
        "whether the reasoning and
answer are logically and factually
correct",
        """Scoring Guide (0-10):
- 10: Fully accurate and logically sound;
no flaws in reasoning or facts.
- 8-9: Mostly correct with minor
inaccuracies or small logical gaps.
- 6-7: Partially correct; some key flaws
or inconsistencies present.
- 4-5: Noticeable incorrect reasoning or
factual errors throughout.
- 1-3: Largely incorrect, misleading, or
illogical.
- 0: Entirely wrong or nonsensical."""
        )
```

```
"logical_coherence": (
        "logical_coherence",
        "whether the reasoning is
internally consistent, clear, and well-
structured",
        """Scoring Guide (0-10):
- 10: Highly logical, clear, and easy to
follow throughout.
- 8-9: Well-structured with minor lapses
in flow or clarity.
- 6-7: Some structure and logic, but a
few confusing or weakly connected parts.
- 4-5: Often disorganized or unclear;
logic is hard to follow.
- 1-3: Poorly structured and incoherent.
- 0: Entirely illogical or
unreadable."""
        )
```

```
"comprehensiveness": (
        "comprehensiveness",
        "whether the thinking considers
all important aspects and is thorough",
        """Scoring Guide (0-10):
- 10: Extremely thorough, covering all
relevant angles and considerations with
depth.
- 8-9: Covers most key aspects clearly
and thoughtfully; only minor omissions.
- 6-7: Covers some important aspects, but
lacks depth or overlooks notable areas.
- 4-5: Touches on a few relevant points,
but overall lacks substance or
completeness.
- 1-3: Sparse or shallow treatment of the
topic; misses most key aspects.
- 0: No comprehensiveness at all;
completely superficial or irrelevant.""" )
```

```
"""---Role---
You are a helpful and rigorous
assistant evaluating the
**{title}** of a generated response.
---Question---
{question}
---Golden Evidences---
{evidences}
---Evaluation Goal---
Evaluate **{goal}** using a **0-10
integer scale**.
{rubric}
---Output Format---
Score (an integer from 0 to 10)
---Generation to be Evaluated---
{response}
"""
```

```
"relevance": (
        "relevance",
        "whether the reasoning and
answer are highly relevant to the
evidence and helpful to the question",
        """Scoring Guide (0-10):
- 10: Fully focused on the evidence;
highly relevant and helpful.
- 8-9: Mostly on point; minor
digressions but overall useful.
- 6-7: Generally relevant, but includes
distractions or less helpful parts.
- 4-5: Limited relevance; much of the
response is off-topic or unhelpful.
- 1-3: Barely related to the evidence or
largely unhelpful.
- 0: Entirely irrelevant."""
        )
```

```
"knowledgeability": (
        "knowledgeability",
        "whether the thinking is rich in
insightful, domain-relevant knowledge",
        """Scoring Guide (0-10):
- 10: Demonstrates exceptional depth and
insight with strong domain-specific
knowledge.
- 8-9: Shows clear domain knowledge with
good insight; mostly accurate and
relevant.
- 6-7: Displays some understanding, but
lacks depth or has notable gaps.
- 4-5: Limited knowledge shown;
understanding is basic or somewhat flawed.
- 1-3: Poor grasp of relevant knowledge;
superficial or mostly incorrect.
- 0: No evidence of meaningful
knowledge.""")
```

```
"factuality": (
        "factuality",
        "whether the reasoning and
answer are based on accurate and
verifiable facts",
        """Scoring Guide (0-10):
- 10: All facts are accurate and
verifiable.
- 8-9: Mostly accurate; only minor
factual issues.
- 6-7: Contains some factual
inaccuracies or unverified claims.
- 4-5: Several significant factual
errors.
- 1-3: Mostly false or misleading.
- 0: Completely fabricated or factually
wrong throughout."""
        )
```

Figure 13: Evaluation prompts of **A-Score** across 3 criteria and **E-Score** across 3 criteria.

# G  LIMITATIONS AND FUTURE DIRECTION

Although GRAPHSEARCH has made progress in advancing GRAPHRAG, there are still some limitations. First, it remains uncertain whether GRAPHSEARCH can unlock greater potential under different training strategies, such as fine-tuning or reinforcement learning. Second, how to integrate it with cutting-edge reasoning models is still an open question. Finally, applying GRAPHSEARCH to scenarios involving multimodal corpora is a direction worthy of further investigation.

# H  THE USE OF LARGE LANGUAGE MODELS (LLMS)

During the completion of this thesis, the scenarios involving the use of LLMs included: using code-completion tools to assist with experiments, and using ChatGPT to polish the draft after the initial writing was completed. LLMs were not involved in any aspects such as the development of research ideas, literature review, and so on.

