# OpenReview forum: "GraphSearch: An Agentic Deep Searching Workflow for Graph Retrieval-Augmented Generation"
_ICLR.cc/2026/Conference — ICLR 2026 Conference Withdrawn Submission_

### Official Review · Reviewer_c9Rn · 2025-10-29

**Soundness:** 2
**Presentation:** 3
**Contribution:** 2
**Rating:** 4
**Confidence:** 3

**Summary:**

This work proposes an deep searching workflow agent accessing both semantic and structural knowledge for complex retrieval-based reasoning tasks. It proposes 6 modules to address the retrieval challenges including shallow retrieval and constrained retrieval scope and conduct comprehensive empirical experiments across 6 benchmarks with ablation studies.

**Strengths:**

- The paper did detailed experiments across multiple benchmarks including both wiki-based datasets and domain-based datasets.
- According to the results, the proposed method remains effective in small LLMs (7B models).
- The paper is well written.

**Weaknesses:**

- The paper has limited contribution in research novel. All six modules the paper proposed such as query decomposition, context refinement, query grounding, query expansion has been extensively studied and applied in prior works. The dual-channel retrieval strategy and the iterative retrieval concept highlighted by the paper is also not a new idea.

- According to the ablation study results reported in Table 3, the LD, EV, and QE modules don't provide significant performance improvement.

- The paper doesn't provide cost / compute comparison between the proposed methods and the baselines since the introduced modules may significantly increases the latency and compute.

**Questions:**

Does the proposed GraphSearch remain effective on large-scale LLMs such as GPT5 / Sonnet4.5 etc.?

---

### Official Review · Reviewer_5r6F · 2025-10-30

**Soundness:** 3
**Presentation:** 2
**Contribution:** 2
**Rating:** 4
**Confidence:** 4

**Summary:**

This paper addresses performance limitations in Graph Retrieval-Augmented Generation (GraphRAG), specifically targeting two key challenges: the reliance on single-round retrieval and insufficient exploitation of structural information within graph data. To overcome these limitations, the authors introduce an agentic search workflow that enables iterative retrieval through multiple components, including query decomposition and context refinement modules. The proposed approach is evaluated on six multi-hop RAG benchmarks and compared against nine baseline methods.

**Strengths:**

1. The problem of iterative retrieval is well-motivated, as the retrieval may make mistakes.
2. The performance improvement is significant when combining this workflow with existing retrieval methods.

**Weaknesses:**

1. The novelty is limited and not well-explained in the paper. (Q1, Q2)
2. The representation of the paper can be further improved, e.g., some figures are not readable. (Q3, Q4, Q5, Q8, Q9)
3. The experiment is not comprehensive, and some implementation details are missing. (Q6, Q7)

**Questions:**

1. While the proposed approach is sound, the paper's novelty remains unclear. Specifically, the components presented in Section 4 appear to be well-established techniques in the broader RAG literature. Even if certain components have not been previously applied to GraphRAG, their adaptation appears straightforward without significant novelty. I recommend the authors clarify the following.
    - First, regarding technical novelty: which components, if any, required non-trivial modifications or novel design choices to work effectively in the GraphRAG setting?
    - Second, concerning graph-specific adaptations: how do your components differ from their counterparts in standard RAG systems, and what unique challenges arise from the graph structure that necessitated these modifications?
    - Third, on contribution positioning: if the main contribution is the integration or combination of existing techniques, please explicitly frame it as such and justify why this particular combination is non-obvious or particularly effective.
2. In Line 76, the connection between GraphSearch components and Limitation (ii) is unclear. The authors claim that their approach addresses this limitation, but I cannot identify which specific components or mechanisms accomplish this. Could the authors explicitly indicate where in the paper (with section references) the solution to Limitation (ii) is described? A clearer mapping between the stated limitations and the proposed solutions would strengthen the paper's clarity.
3. Excessive bold formatting throughout the paper hurts readability. Please significantly reduce its use and let clear writing convey emphasis instead.
4. The current figures may not be suitable for a scientific venue. The cartoon characters are difficult to interpret and may confuse readers unfamiliar with the intended metaphor. I strongly recommend redesigning the figures using conventional technical representations. This would improve both the professional appearance and the communicative effectiveness of the paper.
5. Figure 2 should explicitly show whether the graph is constructed from the text data or exists as an independent database. Please revise the figure to clearly illustrate the relationship between these two data sources.
6. Section 5 evaluates the proposed approach using only a single LLM, which limits the generalizability of the findings. The experiments should include at least one additional model, preferably a more capable one such as GPT or Claude, to demonstrate that the performance gains are robust across different language models rather than specific to one model's characteristics.
7. Section 5.2 does not specify which model serves as the LLM-as-a-Judge evaluator. Please explicitly state the model.
8. The font size in Figure 5 is too small to be legible. Please increase the font size.
9. Figure 8 requires clarification on two points. First, the concept of "relational-channel" applied to text data is unclear. Please explain how this is done. Second, the performance margins between HyperGraphRAG and PathRAG appear minimal. Could you provide statistical significance tests to determine whether these differences are meaningful?

---

### Official Review · Reviewer_nkJW · 2025-10-30

**Soundness:** 2
**Presentation:** 3
**Contribution:** 2
**Rating:** 4
**Confidence:** 4

**Summary:**

This paper proposes **GraphSearch**, a modular agentic workflow for Graph Retrieval-Augmented Generation incorporating six components and dual-channel retrieval (semantic/relational) to address shallow retrieval in existing GraphRAG systems. It achieves deep evidence search through multi-turn "decompose-retrieve-reflect" loops, showing improvements across six benchmarks and seamless integration with various GraphRAG systems (LightRAG, PathRAG, HyperGraphRAG). While the methodology is sound and experiments validate effectiveness, the contribution primarily lies in architectural orchestration rather than novel components, as individual modules largely adapt existing agentic RAG techniques. Key comparisons with baselines (Self-RAG, ReAct) are incomplete, and analysis of computational cost-performance trade-offs is lacking. The paper offers significant engineering value but limited conceptual innovation.

**Strengths:**

1. The pape applies agentic workflows to Graph Retrieval-Augmented Generation (GraphRAG), addressing the practical limitation of insufficient retrieval depth in existing systems. This application is itself significant, providing a systematic, modular six-component framework for deep graph searching.

2. The dual-channel retrieval strategy introduced by GraphSearch is innovatively designed to align with the nature of a Graph Knowledge Base. Retrieval is executed by simultaneously utilizing semantic queries (targeting text/node content) and relational queries (targeting graph structure/edges), which more effectively harnesses the value of graph data. Furthermore, the discovered modality-function alignment property (Figure 8) provides evidence for this design.

3. The study features a comprehensive experimental evaluation across six different and diverse benchmark datasets. The results are compelling, demonstrating consistent and significant improvements over single-pass GraphRAG methods. Moreover, it proves its plug-and-play capability across three different GraphRAG systems (LightRAG, PathRAG, HyperGraphRAG), which greatly increases its practical impact.

4. The paper exhibits good clarity, explicitly articulating the problem with motivating examples (Figure 1). The methodology section is well-structured and provides proper mathematical formalization. The analysis of retrieval quality evolution (Figure 7) offers valuable insights, and the detailed appendix information (such as prompt templates and case studies) ensures the research's reproducibility.

5. The framework addresses an important bottleneck in GraphRAG systems. Its generalizability and applicability across different implementations enhance its practical impact. Furthermore, its utility is particularly evident in overcoming limitations related to retrieval depth.

**Weaknesses:**

1. The work primarily integrates existing techniques (query decomposition, iterative retrieval, reflection) from the broader agentic RAG literature. While the orchestration for GraphRAG is valuable, the individual modules lack fundamental innovation. For example, query decomposition is seen in (Cao et al., 2023) and iterative retrieval with reflection resembles Self-RAG (Asai et al., 2024). The authors must better articulate the specific novelty beyond the application domain.

2. Firstly, ReAct results are insufficient. ReAct records the results of LLM inference, but ReAct itself can be combined with Naive RAG and other GraphRAG Baselines. Secondly, there is no comparison with other established agentic RAG methods (e.g., Self-RAG, PlanRAG, Search-o1). This omission is critical, as the paper's core claim is about the superiority of agentic interaction. Without these comparisons, it is unclear whether the performance gains stem from the graph-specific design or merely from the known benefit that simple iterative querying can significantly boost the accuracy of multi-hop QA in RAG systems.

3. Multi-turn agentic interactions inherently increase latency and API costs. The paper lacks analysis of key metrics: number of iterations required, total token consumption, and wall-clock time. While higher accuracy might justify the added overhead in certain scenarios, the paper is critically missing a cost-benefit analysis to demonstrate how much performance improvement is necessary to warrant the additional computational expense.

4. Some problems in Ablation Study. The paper contains a minor error by labelling the Logic Drafting (LD) module as "AD" in the results table and the ablation results are in Table 3 rather than Appendix A. Crucially, the transition from [QD, CR, QG] to [QD, CR, QG, LD] shows *minimal or negative* improvement, yet the paper fails to explain this counterintuitive result, raising serious questions about the necessity of the Logic Drafting module.

5. The experiments on Wikipedia datasets (HotpotQA/MuSiQue/2Wiki) only use 300 resampled examples per set, raising questions about scalability. Since the original datasets are larger (e.g., HipporRAG2's sampled datasets contain 1000 examples), the choice to further restrict the sample size needs clarification.

**Questions:**

1. Please provide a detailed analysis of the computational costs of GraphSearch compared to single-pass baselines, including metrics such as the average number of iterations, total token consumption, and wall-clock time. Since agentic methods incur significantly higher overhead, what is the explicit cost-efficiency trade-off? At what point does the performance improvement justify the increased computational expense?

2. The paper's claim of agentic superiority is significantly weakened by incomplete comparisons. Please provide comprehensive results against key, established agentic RAG methods, such as Self-RAG and PlanRAG. Furthermore, the evaluation of ReAct (specifically when integrated with an RAG system) is incomplete; please provide the full set of evaluation metrics for a fair comparison against the proposed agentic workflow.

3. The Logic Drafting (LD) module is presented as a component of the novel framework, yet Table 3 shows its marginal contribution is inconsistent or even negative in some settings. Is the LD module truly necessary? Please explain the reason for this counterintuitive result.

4. The finding that limiting each channel to its aligned modality (semantic-text, relational-graph) performs comparably to using all data seems highly counterintuitive. Please provide a deeper theoretical or empirical explanation for this key "modality-function alignment" property and why it leads to such results.

5. The experiments on the Wikipedia-based datasets (HotpotQA/MuSiQue/2Wiki) only use 300 resampled examples, which is smaller than the 1,000-sample subsets often used (e.g., in HippoRAG). Please clarify the rationale for this further reduction in sample size: Was this choice made purely due to computational constraints? Furthermore, please describe the exact sampling methodology used (e.g., was it random sampling?) to allow reviewers to assess the representativeness of the reported results and their generalizability to larger scales.

---

### Official Review · Reviewer_U8Mb · 2025-10-31

**Soundness:** 3
**Presentation:** 3
**Contribution:** 2
**Rating:** 2
**Confidence:** 4

**Summary:**

This paper proposes GraphSearch, an agentic pipeline for GraphRAG. It iteratively retrieves from the graph KBs and splits the retrieval into two channels: the semantic & relational channels. In this way, GraphSearch could mitigate the shallow retrieval issue and better exploit structural data. GraphSearch includes six manually designed modules and answers only when enough evidence is gathered. On six multi-hop RAG benchmarks, GraphSearch shows improvement over one-shot GraphRAG pipelines.

**Strengths:**

- This paper is well-motivated, which critically points out the two key issues of current GraphRAG methods
- The overall design makes sense, and should help mitigate the two spotted issues.
    - The two-channel design is especially interesting.
- Experiments show good improvements.

**Weaknesses:**

- The pipeline echoes known agentic design, e.g., ReAct/IRCoT. The technical contribution could be somewhat limited.
- The design with six modules makes the system complex, and also limits the contributions largely systemic, not algorithmic.

**Questions:**

See above.

---

### Note · Authors · 2025-12-20

I have read and agree with the venue's withdrawal policy on behalf of myself and my co-authors.